# Development of Taccalonolide AJ-Hydroxypropyl-β-Cyclodextrin Inclusion Complexes for Treatment of Clear Cell Renal-Cell Carcinoma

**DOI:** 10.3390/molecules25235586

**Published:** 2020-11-27

**Authors:** Jing Han, Siwang Zhang, Junxin Niu, Chunli Zhang, Weichen Dai, Yuanyuan Wu, Lihong Hu

**Affiliations:** 1Jiangsu Key Laboratory for Functional Substance of Chinese Medicine, Jiangsu Collaborative Innovation Center of Chinese Medicinal Resources Industrialization, State Key Laboratory Cultivation Base for TCM Quality and Efficacy, Jiangsu Key Laboratory for Pharmacology and Safety Evaluation of Chinese Materia Medical, Nanjing University of Chinese Medicine, Nanjing 210023, China; Jiangniao172@126.com (J.H.); 15951878760@163.com (S.Z.); zhangqian0823@126.com (C.Z.); daiweichen1988@163.com (W.D.); 2State Key Laboratory of Drug Research, Shanghai Institute of Material Medical, Chinese Academy of Sciences, 501 Haike Road, Shanghai 201203, China; qiusisiqiu3773@126.com

**Keywords:** taccalonolide AJ, microtubule stabilizer, hydroxypropyl-β-cyclodextrin, inclusion complex, clear-cell renal-cell carcinoma

## Abstract

Background: Microtubule-targeted drugs are the most effective drugs for adult patients with certain solid tumors. Taccalonolide AJ (AJ) can stabilize tubulin polymerization by covalently binding to β-tubulin, which enables it to play a role in the treatment of tumors. However, its clinical applications are largely limited by low water solubility, chemical instability in water, and a narrow therapeutic window. Clear-cell renal-cell carcinoma (cc RCC) accounts for approximately 70% of RCC cases and is prone to resistance to particularly targeted therapy drugs. Methods: we prepared a water-soluble cyclodextrin-based carrier to serve as an effective treatment for cc RCC. Results: Compared with AJ, taccalonolide AJ-hydroxypropyl-β-cyclodextrin (AJ-HP-β-CD) exhibited superior selectivity and activity toward the cc RCC cell line 786-O vs. normal kidney cells by inducing apoptosis and cell cycle arrest and inhibiting migration and invasion of tumor cells in vitro. According to acute toxicity testing, the maximum tolerated dose (MTD) of AJ-HP-β-CD was 10.71 mg/kg, which was 20 times greater than that of AJ. Assessment of weight changes showed that mouse body weight recovered over 7–8 days, and the toxicity could be greatly reduced by adjusting the injections from once every three days to once per week. In addition, we inoculated 786-O cells to generate xenografted mice to evaluate the anti-tumor activity of AJ-HP-β-CD in vivo and found that AJ-HP-β-CD had a better tumor inhibitory effect than that of docetaxel and sunitinib in terms of tumor growth and endpoint tumor weight. These results indicated that cyclodextrin inclusion greatly increased the anti-tumor therapeutic window of AJ. Conclusions: the AJ-HP-β-CD complex developed in this study may prove to be a novel tubulin stabilizer for the treatment of cc RCC. In addition, this drug delivery system may broaden the horizon in the translational study of other chemotherapeutic drugs.

## 1. Introduction

Taccalonolides are plant-derived natural steroids isolated from the genus *Tacca* with novel microtubule-stabilizing activity. The most abundant of these steroids is taccalonolide A (0.1%), and it was first isolated in 1987 [1] and identified as a microtubule stabilizer in 2003 [2]. Compared to other microtubule stabilizers, taccalonolides can circumvent multidrug resistance, such as mutations in the taxane-binding site, the overexpression of multidrug resistance protein 7, P-glycoprotein, and the βIII tubulin isotype in vitro [3,4,5]. Most importantly, taccalonolides show excellent anti-tumor activity in paclitaxel- and doxorubicin-resistant murine tumor models [6]. However, there have been no natural taccalonolides detected that directly interact with microtubules or enhance purified tubulin polymerization.

Semi-synthesized taccalonolide AJ (AJ) [7] was the first of the taccalonolides that was found to directly interact with purified tubulin, which occurred by covalent binding of the C-22,23 epoxide moiety to β-tubulin D226 [8]. Compared to paclitaxel, AJ exhibited a half-maximal inhibitory concentration value of 4.2 nM [7]. These properties make AJ attractive next-generation microtubule-stabilizing agents (MSAs) for chemotherapy. However, AJ showed no indication of antitumor effects and a narrow therapeutic window (maximum tolerated dose (MTD) = 0.5 mg/kg). Covalent binding to β-tubulin, a high frequency of injection (once every 2 days), and an unsuitable vehicle (5% EtOH: 95% phosphate-buffered saline (PBS)) may play important roles [6]. The highly oxygenated pentacyclic steroid structure contributed to the poor water solubility. The poor water instability of AJ was due to the activity group C-22,23 epoxide, which induced a β configuration; this configuration inactivates hydrolysis in water. Therefore, the narrow therapeutic window and poor water solubility and instability seriously limit AJ clinical applications.

Cyclodextrins may readily form inclusion complexes with hydrophobic lumens to encapsulate insoluble drugs or to design and assemble different supramolecular structures, such as micelles, vesicles, and nanoparticles, to increase the solubility, stability, and bioavailability of drugs, which are widely used in the field of medicine [9,10,11]. The water solubility and anti-tumor activity of paclitaxel (PTX) can be improved by the inclusion of different supramolecular macrorings, such as methylated β-cyclodextrin (β-CD) [12], hydrocarbon chain modified β-CD derivatives [13], hydroxypropyl-β-CD (HP-β-CD) [14], and β-CD [15].

Clear-cell renal-cell carcinoma (cc RCC) accounts for approximately 70% of renal cell carcinoma (RCC) cases and displays intra-tumor heterogeneity [16]. The current medical treatments for RCC include chemotherapy, radiotherapy, targeted therapy, and immunotherapy [17]. PTX, an effective MSA and apoptosis inducer, is widely used in chemotherapy for multiple cancers, such as cc RCC [18,19]. Sunitinib, an oral tyrosine kinase inhibitor, is approved for cc RCC treatment [20]. However, 20–40% of patients eventually relapse due to the development of resistance to these drugs [21]. Therefore, there is an urgent need to seek more effective novel therapies.

In this study, we chose a taccalonolide AJ-hydroxypropyl-β-cyclodextrin (AJ-HP-β-CD) inclusion and found that AJ-HP-β-CD had superior solubility and stability and showed stronger selectivity on the cc RCC cell line 786-O vs. normal kidney and anti-tumor activity than AJ. Based on its robust cytotoxicity, we further evaluated the effect of AJ-HP-β-CD on 786-O cell growth and explored the underlying mechanism both in vivo and in vitro models.

## 2. Results

### 2.1. Optimization, Preparation, and Characterization of AJ-HP-β-CD Inclusion Complexes

#### 2.1.1. Optimization of Cyclodextrins for AJ-HP-β-CD Inclusion Complexes

AJ (Appendix A) was hydrolyzed in an aqueous solution, and AJ-D was the main hydrolysis product (Figure 1A). Cyclodextrins (Appendix A) can improve the solubility and stability of insoluble drugs, and the inclusion complexes of AJ were optimized by the response surface method with different cyclodextrins (HP-α-CD, HP-β-CD, HP-γ-CD, and Sulfobutylether-β-CD). According to the solubility of the inclusion complex, we chose HP-β-CD as a suitable type of cyclodextrin (Appendix A).

#### 2.1.2. Preparation of AJ-HP-β-CD Inclusion Complexes

The experimental results showed that the phase solubility of AJ in the aqueous solution of HP-β-CD presented a linear relationship, which was of type AL, indicating that a 1:1 molar ratio was formed between AJ and HP-β-CD, and the addition of HP-β-CD significantly increased the solubility of AJ in aqueous solution (Figure 1B).

As shown in Figure 1C, the encapsulation rate was significantly related to the feed ratio, which improved with the increase in the input amount of HP-β-CD. When the molar ratio of AJ and HP-β-CD was 1:1, the encapsulation rate of the cladding compound was the highest, and then the encapsulation rate decreased.

#### 2.1.3. Characterization of AJ-HP-β-CD Inclusion Complexes

AJ + HP-β-CD (Appendix A) is a mixture of AJ and HP-β-CD, which is fully mixed at a molar ratio of 1:1. AJ and HP-β-CD were simple superpositions of the infrared spectra of the two substances, indicating that no other interaction occurs during the physical mixing process. The infrared spectra of AJ-HP-β-CD (Appendix A) were similar to those of HP-β-CD, indicating that HP-β-CD almost covered the characteristic peaks of AJ in the range of 800 cm^−1^ to 1600 cm^−1^ and no other new peaks appeared, suggesting that the bonding process was only due to some weak interactions (Figure 1D).

AJ is a kind of plate aggregate crystal with different sizes, and the surface morphology of HP-β-CD appears as an amorphous shrinkage circular particle. As seen from the photos of the physical mixture, AJ crystals adhered to the smooth surface of HP-β-CD, and there was no interaction between AJ and HP-β-CD because the original morphology of each component remained unchanged. However, AJ-HP-β-CD showed a uniform amorphous structure without the original morphology of AJ and HP-β-CD, indicating that AJ and HP-β-CD reacted with each other and successfully formed AJ-HP-β-CD (Figure 1E). All the data indicated that it is feasible to combine AJ with HP-β-CD at a molar ratio of 1:1, and this kind of encapsulation could greatly improve the solubility of AJ.

### 2.2. The Stability of AJ-HP-β-CD

AJ (5% EtOH: 95% PBS) started hydrolyzing at approximately 6 h, but the activity faded nearly 50% after 24 h; however, AJ-HP-β-CD showed no significant change after 72 h in the water and maintained similar stability in anhydrous ethanol (Figure 2A,B).

### 2.3. In Vitro Cytotoxicity Study of AJ-HP-β-CD

The effect of Taccalonolide B (TB) AJ, PTX, and AJ-HP-β-CD on cell proliferation was detected by MTS (3-(4,5-dimethylthiazol-2-yl)-5-(3-carboxymethoxyphenyl)-2-(4-sulfophenyl)-2H-tetrazolium) assay. The data showed that compared with TB and AJ, AJ-HP-β-CD displayed robust cytotoxic activity against ten of the tested cancer cell lines: A2780, 786-O, ACHN, HepG2, Hep-3B, Huh-7, HCT116, MCF-7, normal HEK293 cells, and HUVECs (Table 1). Interestingly, the IC50 of AJ and AJ-HP-β-CD in primary cc RCC 786-O cells (16.01 ± 1.54, 3.51 ± 0.79) was less than 50% of that of metastatic papillary RCC ACHN (35.96 ± 1.08, 9.82 ± 1.02), indicating that AJ and AJ-HP-β-CD had higher cytotoxic activity on primary cc RCC. Compared with a single AJ. The IC_50_ of AJ-HP-β-CD (3.51 ± 0.79) was approximately one-quarter lower than that of AJ (16.01 ± 1.54). Therefore, AJ-HP-β-CD may have more advantages in the treatment of cc RCC.

### 2.4. AJ-HP-β-CD Inhibits 786-O Cell Invasion and Migration In Vitro

The cytotoxicity of AJ-HP-β-CD toward 786-O cells was detected by CCK-8 assay. Compared with the vehicle group, the PTX, AJ, and AJ-HP-β-CD (1, 3, 10, 30 nM) groups had significantly reduced cell proliferation (both *p* < 0.01) (Figure 3A). AJ-HP-β-CD in the 10 and 30 nM inhibitors 786-O cells proliferation significantly compared with the AJ groups at 24, 48, and 72 h (both *p* < 0.01).

After 24 h of treatment with PTX, AJ, and AJ-HP-β-CD (1, 3, 10 nM), cell invasiveness decreased significantly compared with that in the vehicle group (both *p* < 0.05) (Figure 3B,C). AJ-HP-β-CD at 10 nM significantly inhibited cell invasion compared with the AJ group (*p* < 0.05).

PTX, AJ, and AJ-HP-β-CD (1, 3 10 nM) groups inhibited 786-O cell migration (*p* < 0.05) (Figure 3D,E). The migration ability of AJ-HP-β-CD at 10 nM was significantly stronger than that of AJ (*p* < 0.01) (Figure 3D,E). All these findings suggest that the inhibited migration and invasion ability of AJ were improved after inclusion with AJ-HP-β-CD.

### 2.5. AJ-HP-β-CD Promotes Microtubule Accumulation in 786-O Cells

The effect of AJ-HP-β-CD treatment on cell microtubules was examined by fluorescence microscopy following immunofluorescent staining of tubulin. As microtubule-stabilizing agents, PTX-treated cells appear to form bundles of microtubules (Figure 4). Cells treated with AJ exhibited bundles of microtubules, and these microtubule bundles seem to be quite shorter in length (Figure 4). The microtubule in AJ-HP-β-CD treated cells consistently appear to fill more of the cytoplasm and seem to be quite short in length (Figure 4).

### 2.6. AJ-HP-β-CD Arrests 786-O Cells in G2/M Phase and Induces Cell Apoptosis

To explore the effect of AJ-HP-β-CD on the cell cycle and apoptosis, we performed flow cytometry of the 786-O cells after treated with PTX (10 nM), AJ (10 nM), or AJ-HP-β-CD (1, 3, 10 nM). The data showed that 1 and 10 nM AJ-HP-β-CD treatment for 12 h was sufficient to arrest 786-O cells in G2/M phase (Figure 5A,C). Apoptotic cells were detected by Annexin V/PI (Propidium Iodide)-labeled flow cytometry, and after exposure of 786-O cells to drugs for 24h, an increased number of apoptotic cells were detected in AJ (5.91 ± 0.99%) treatment. The percentage of apoptotic cells was 12.30 ± 0.25% after 3 nM AJ-HP-β-CD treatment and gradually increased up to 16.07 ± 1.1% at a dose of 10 nM (Figure 5B,D).

### 2.7. AJ-HP-β-CD Inhibits 786-O Xenograft Growth in Mice

The acute toxicity testing showed less weight loss and toxicity in AJ-HP-β-CD at different doses for 2 weeks. According to the changes in mouse body weight within 2 weeks after different doses of AJ-HP-β-CD (Appendix A), the body weight decreased within 1–6 days and increased after 6–8 days, suggesting that the administration cycle of AJ- HP-β-CD should be adjusted once a week, which can greatly reduce toxicity.

The tissue distribution data showed that compared with AJ (2 mg/kg, 1283.33 ± 98.91 ng/g, and 19.33 ± 3.51 ng/g), AJ-HP-β-CD (2 mg/kg) administered by gavage led to higher kidney deposition (2016.67 ± 156.15 ng/g and 116.67 ± 10.79 ng/g, *p* < 0.05) after 5 min and 30 min (Appendix A). compared with AJ-HP-β-CD gavage, AJ-HP-β-CD intravenously (2 mg/kg) had much higher kidney deposition (2804.33 ± 196.5 μg/g, *p*< 0.05) after 5 min and 30 min (Appendix A), respectively. These data suggested that intravenously delivered AJ-HP-β-CD had a more active pharmaceutical profile for cc RCC treatment than AJ.

We studied the tumor-suppressive effect of AJ-HP-β-CD in nude mice xenografted with 786-O cell-derived engraftment. The results showed that the tumor volume was reduced and tumor growth was inhibited significantly in the docetaxel, sunitinib, and AJ-HP-β-CD groups compared with vehicle control animals (Figure 6A,B, Table 2). The in vivo pharmacodynamics results showed that sunitinib had the best effect of inhibiting tumor growth (the tumor volume was reduced by approximately 28.71%), which was followed by docetaxel and AJ-HP-β-CD (Figure 6A–D). Compared with docetaxel (10.0 mg/kg/day) [6] and sunitinib (80 mg/kg/day for 1st week, 40 mg/kg/day for last two weeks) [22], AJ-HP-β-CD showed a low dosage (1.0 or 2.0 mg/kg) and long interval (once a week) during the 3-week treatment period significantly and dose-dependently reduced the tumor volume (Figure 6A,B, Table 2) and tumor growth (Figure 6C,D) with less weight loss or other toxicities. In addition, no significant change in body weight was observed in the AJ-HP-β-CD groups compared with the control, docetaxel, and sunitinib groups (Figure 6E,F). The sunitinib and docetaxel groups exhibited weight loss (Figure 6E,F), and the animals showed weakness and inactivity as well as the production of yellowish urine.

All these data indicate that low-dose AJ-HP-β-CD can dramatically reduce tumor growth with minimal adverse effects, expanding the therapeutic window, and may, therefore, prove to be a promising drug for treating kidney cancer.

## 3. Discussion

Metastasis and drug resistance constitute two major barriers in the clinical treatment of RCC patients [23]. As a major component of the cytoskeleton, microtubules consist of αβ-tubulin heterodimers and have been recognized as attractive targets for cancer chemotherapy [8]. Semi-synthesized taccalonolide AJ (AJ) is a microtubule inhibitor, but unlike paclitaxel, it can overcome multidrug resistance. Risinger et al. reported that taccalonolides were effective both in vitro against cell lines that overexpressed P-glycoprotein, a multidrug resistance protein, and in vivo against paclitaxel-resistant tumors [6]. The difference was elaborated by the same group later that AJ could target interphase microtubules at concentrations comparable to their anti-proliferative effects, while paclitaxel needed a more than 30-fold higher concentration to target microtubule bundling compared with IC_50_. These results indicated that AJ was a potent drug for cc RCC.

Knowing that the small therapeutic window and low aqueous solubility of AJ seriously limit its clinical applications, we developed an AJ cyclodextrin-based carrier (AJ-HP-β-CD) in the present study. AJ-HP-β-CD was prepared in accordance with a molar ratio of 1:1, and its water solubility and stability were improved by liquid phase detection. Plasma pharmacodynamics showed that AJ-HP-β-CD was greatly improved (Appendix A) [24] and its enrichment time in the kidney was longer than that of AJ (Appendix A). Moreover, because AJ is a new covalent microtubule stabilizer, frequent administration will lead to increased toxicity and a small therapeutic window. According to the weight regained in approximately one week, its MTD had increased nearly 20 times by changing once every three days to once a week. Due to the high water solubility and low toxicity of HP-β-CD [13], various functional groups may be attached to cyclodextrins to modify their chemical properties. Therefore, AJ will be a valuable drug with low toxicity, a larger therapeutic window, and better aqueous solubility and stability in cyclodextrin complexes.

Through in vitro tumor spectrum screening, we found that the treatment effect of AJ-HP-β-CD on cc RCC was selective, and its IC_50_ was changed from the original 16.01 ± 1.54 nM to 3.51 ± 0.79 nM. Through in vitro cloning experiments and lateral migration experiments, it was found that the activity of AJ-HP-β-CD was greatly improved, and the inhibitory effect of 1 nM AJ-HP-β-CD on 786-O cells was basically similar to that of AJ at 10 nM. Flow cytometry analysis revealed that the AJ-HP-β-CD antigonid was sufficient to arrest 786-O cells in G2/M phase. All the results showed that the anti-786-O activity of AJ was increased by approximately 10 times after the inclusion of cyclodextrin.

The in vivo pharmacodynamics results showed that sunitinib had the best effect of inhibiting tumor growth, followed by AJ-HP-β-CD and docetaxel. Compared with docetaxel (10.0 mg/kg/day) and sunitinib (80 mg/kg/day), AJ-HP-β-CD showed a low dosage (1.0 or 2.0 mg/kg) and long interval (once a week) during the 3-week treatment period with less weight loss or other toxicities. In addition, no significant change in body weight was observed in the AJ-HP-β-CD group compared with the docetaxel and sunitinib groups. All these data indicate that low-dose taccalonolides can dramatically reduce tumor growth with minimal adverse effects and may, therefore, prove to be a promising drug for kidney cancer.

## 4. Materials and Methods

### 4.1. Materials and Animals

The analysis was carried out with an Agilent 1260 HPLC system (Agilent, Waldbronn, Germany), and detection was performed with an Evaporative Light Scattering Detector (ELSD) (Agilent, Waldbronn, Germany). Ultra-pure water was prepared with a Milli-Q water system (Bedford, MA, USA). Hydroxypropyl-β-cyclodextrin (HP-β-CD) was provided by Beyotime (Shanghai, China). Acetonitrile and methanol were of chromatographic grade (Merck, Darmstadt, Germany). All other reagents were of analytical grade.

### 4.2. Detection Method and Content Standard Curve of AJ

AJ was synthesized from taccalonolide A (TA) following our previous procedures [4]. The ELSD Waters 2695 system (Waters, Milford, MA, USA) was used for HPLC analysis. The HPLC column used was an X-bridge C18 column (internal diameter 250 mm × 4.6 mm, 5 μm) (Waters, Dublin, Ireland). The mobile phase was acetonitrile and water (containing 0.5 ‰ methanoic acid) with gradient elution (20:80~80:20, *v*/*v*, 0–20 min) at a flow rate of 0.5 mL/min. Following HPLC separation, the amount of AJ was measured by the peak-area ratio method.

To determine the standard calibration curve, serial dilutions of AJ in methanol were prepared (1, 0.5, 0.25, 0.125, 0.064, 0.032, and 0.016 mg/mL). The area under the curve (AUC) in the HPLC chromatograms was calculated for different AJ concentrations, and the standard curve was obtained by linear regression analysis. A linear regression equation was determined (r^2^ > 0.99) and used to calculate the AJ amount in subsequent experiments (Appendix A).

### 4.3. Phase Solubility and Inclusion Rate Study of the AJ-HP-β-CD Complex

A phase solubility study was carried out in distilled water to examine AJ and AJ-HP-β-CD solubility in water [25]. An excess amount of AJ was mixed with increasing concentrations of AJ-HP-β-CD (0,2.5,5,7.5,10, 12.5 molar equivalent). After equilibration, undissolved AJ was removed from the suspension by filtration using a 0.45 μm membrane filter (Acrodisc^®^ Syringe Filters, Nanjing, China). The concentration of AJ-HP-β-CD was determined by HPLC in triplicate.

AJ (1 mM) was accurately weighed, and AJ and AJ-HP-β-CD had a feed ratio of 2:1,1:1,1:2,1:3,1:4. Measurements were performed, and the inclusion rate was detected by the standard curve method according to 3.2.

### 4.4. Preparation of the AJ-HP-β-CD Complex

Preparation of AJ-HP-β-CD inclusion complexes was performed as follows: Different molar ratios of AJ-HP-β-CD (1:1) were weighed. AJ was initially dissolved in 5 mL ethanol and then slowly added into the AJ/HPβCD solution in deionized water (20 mL). The resulting mixture was heated to reflux with stirring until all solids were dissolved. After removing the volatiles on the rotavapor, the residue was lyophilized to give a white power, which was stored in a moisture-proof container for subsequent experiments.

### 4.5. The Stability of AJ-HP-β-CD and AJ

AJ (1 mg) and AJ-HP-β-CD (1 mg) were accurately weighed and dissolved in EtOH, 5% EtOH: 95% PBS, and AJ-HP-β-CD (1 mg AJ) in water at room temperature. The amount of AJ was measured at 1, 2, 3, 4, 5, 6, 7, 8, 9, 10, 11, 12, 18, 24, 30, 42, 48, 60, and 72 h, and the concentration of AJ-HP-β-CD was determined by HPLC in triplicate.

### 4.6. Material Characterization of the AJ-HP-β-CD Complex

Precisely weighing the right amount of AJ, AJ + HP-β-CD (AJ and HPβCD were thoroughly mixed at a molar ratio of 1:1), HP-β-CD, and AJ-HP-β-CD, which were ground and mixed with potassium bromide (KBr), Fourier transform infrared spectroscopy of the disks was scanned using an FT-IR spectrometer (Nicolet iN10, Bruker, Thermo, Waltham, MA, US). Scanning electron microscopy (SEM) was conducted on a FIB-SEM (Focused ion beam-SEM) microscope (Crossbeam 340, Zeiss, Jena, Germany).

### 4.7. Cell Lines and Cell Culture

The human ovarian cancer cell line A2780, RCC cell lines 786-O and ACHN, and hepatocellular carcinoma (HCC) cell lines HepG2, Hep-3B, Huh-7, HCT116, and MCF-7 were purchased from the Cell Culture Center at the Institute of Basic Medical Sciences of the Chinese Academy of Medical Sciences (Shanghai, China). All cells were cultured in RPMI-1640 medium (Gibco, Grand Island, NY, USA) supplemented with 10% fetal calf serum (Gibco, Grand Island, NY, USA) and 1% penicillin-streptomycin (Hy Clone, Thermo, Guangzhou, China) at 37 °C in an atmosphere of 95% O_2_ and 5% CO_2_.

### 4.8. Cell Proliferation Assay

An CCK-8 assay kit (Beyotime, Shanghai, China) was used to determine cell proliferation according to the manufacturer’s instructions. Cells were plated at 1 × 10^4^ cells/mL in a 96-well plate, and 150 mL was placed in each well. TB, AJ, and PTX were dissolved in DMSO, and AJ-HP-β-CD was dissolved in saline. The vehicle (DMSO) was used as a control in all experiments at a maximum concentration of 0.1%. After 24, 48, or 72 h of treatment, CCK-8 was added to the culture medium. After 4 h of incubation, the OD490 was determined. The IC_50_ was calculated with GraphPad Prism 8.0. All experiments were repeated in triplicate.

### 4.9. Cell Invasion and Migration Assays

To analyze the invasion and migration ability of cells, scratch tests and transwell migration assays were performed according to published methods. For the cell scratch test [26], cells were digested into a single cell suspension, inoculated into a 6-well plate at a concentration of 10^6^ cells/well overnight, and scratched vertically with a 100 μL micropipette tip the next day. Thereafter, the cells were washed twice with PBS and placed in a serum-free culture medium. After 12 h, cells were counted under an inverted phase-contrast microscope in five random fields. Each migration test was run in triplicate. The wound area was photographed immediately at 0 (W0) and 12 h (W12) and the relative invaded rate in each group was calculated as the following formula (1):(1)$$\mathrm{Relative}\;\mathrm{invaded}\;\mathrm{rate}\;\left(\%\right)=\frac{W0-W12}{W0}\times 100\%$$

Transwell migration assays [27] were performed according to standard protocols. 786-O cells (1 × 10^5^ cells/mL in 0.2 mL) were seeded in each chamber insert (8 μm pore size, Corning) of a 12-well plate. After 12 h of incubation, cells in the transwell chamber were fixed with 4% formaldehyde in 1× PBS and stained with 3% crystal violet staining buffer. The fixed crystal violet in each chamber insert was dissolved in 500 μL of 2% acetic acid. Absorbance (A) of the cleanout fluid was determined using a microplate reader (Infinite M1000, Tecan, Switzerland) at 570 nm.

### 4.10. Immunofluorescence Assays

786-O cells were seeded at 5 × 10^3^ cells/well in a 24-well plate and cultured overnight. After 24 h of treatment, cells were fixed with precooled 4% formaldehyde in phosphate-buffered saline (PBS) for 15 min at 4 °C, permeabilized in 0.1% Triton X-100 for 20 min, incubated with an anti–β-tubulin antibody (1:200, Abcam, MO, USA) at 4 °C overnight, reintubated with a FITC (Fluorescein Isothiocyanate)-conjugated secondary antibody, and then contained with DAPI (4′,6-Diamidino-2-Phenylindole, 1:500) for 5 min before being observed under a fluorescence microscope (Laika, Berlin, Germany). Representative images were taken of cellular microtubules [28].

### 4.11. Cell Cycle Distribution and Apoptosis Analysis

The cell cycle analysis was performed with a cell cycle detection kit (KeyGen Biotech, Nanjing, China). Cells were seeded into 6-well plates and incubated overnight. After treatment with the test substances for 12 h, cells were harvested, fixed in cold 70% ethanol overnight at −20 °C, washed with PBS, and stained with a PI solution (20 mg/mL PI and 20 mg/mL RNAase in PBS) for 30 min. The cell cycle was analyzed using FACS Cytometry (San Jose, CA, USA) [29].

Apoptosis was detected using an annexin V-FITC/PI apoptosis detection kit (BD Biosciences Pharmingen, San Diego, CA, USA). 786-O cells were seeded at a density of 5×10^5^ per well in six-well plates and cultured with different concentrations of PTX (10 nM) and AJ-HP-β-CD (1, 3, and 10 nM) for 24 h. A total of 1 × 10^6^ to 3 × 10^6^ cells were washed with ice-cold PBS and resuspended in 1× binding buffer (10 mM Hepes/NaOH (pH 7.4), 140 mM NaCl, and 2.5 mM CaCl_2_) at a concentration of 1 × 10^6^ cells/mL. Five microliterss of annexin V-FITC solution (25 μg/mL) and 5 μL of dissolved PI (250 μg/mL) were added to 100 μL of the cell suspension. Cells were then gently vortexed and incubated at room temperature in the dark for 15 min. Then, 400 μL of ice-cold binding buffer was added, mixed gently, and analyzed by flow cytometry within 1 h (FACS Calibur, Becton Dickinson, San Jose, CA, USA) [29].

### 4.12. Xenografts

Four-week-old BALB/c nude mice were purchased from Beijing HFK Bioscience Co., Ltd. (Beijing, China). The animals were housed in a light- and temperature-controlled room (21–22 °C; relative humidity 60–65%) and maintained on a standard diet and water. All animal studies complied with the ARRIVE (Animal Research: Reporting of In Vivo Experiments) guidelines, and all the experimental protocols were approved by the Ethics Committee of Animal Experiments of Nanjing University of Chinese Medicine (SYXK(S)2018-0049, Nanjing, China). The human RCC cell line 786-O was obtained from the Shanghai Institute of Biochemistry and Cell Biology (Shanghai, China). A small piece of tumor approximately 1 mm^3^ was implanted into the right flank of each male experimental athymic nude mouse. Mice were administered the designated drugs when the tumor size reached an average of 250 mm^3^ (100–300 mm^3^ range).

The mice tail were intravenously administered weekly with 200 μL (the Control received an equivalent volume of saline) of AJ-HP-β-CD at two different concentrations (1.0 mg/kg or 2.0 mg/kg), ocetaxel group mice tail intravenously treated with docetaxel (MSA drug control) at 10.0 mg/kg weekly. Sunitinib group mice received an oral administration of sunitinib (Tyrosine Kinase Inhibitor, TKI positive control) 80 mg/kg at 1 week and 40 mg/kg at 2–3 weeks. The drugs were started on day 22 after tumor cell engraftment. Mouse body weight and tumor size were measured twice a week. The tumor volume was calculated using the following Equation (2):Tumour Volume (mm^3^) = length (mm) × width (mm) × height (mm)(2)

The tumor-inhibiting rate was calculated using the following Equation (3):(3)$$\mathrm{Tumour}\;\mathrm{Inhiting}\;\mathrm{Rate}\;\left(\%\right)=\frac{\mathrm{tumor}\;\mathrm{weight}\;\mathrm{in}\;\mathrm{treatment}\;\mathrm{group}}{\mathrm{tumor}\;\mathrm{weight}\;\mathrm{in}\;\mathrm{control}\;\mathrm{gropu}}\times 100\%$$

### 4.13. Statistical Analysis

Data are expressed as the mean ± standard deviation (SD). Figures were processed by GraphPad Prism 8.0 (GraphPad Software, San Diego, CA, USA). Statistical analysis was calculated using two-way ANOVA with Tukey’s post hoc test and Student’s *t*-test, and *p* < 0.05 was considered statistically significant.

## 5. Conclusions

Our study demonstrates that AJ-HP-β-CD improves the therapeutic window, water solubility, and stability of AJ. AJ-HP-β-CD can arrest cc RCC in G2/M phase and induce cell apoptosis by disrupting microtubule dynamics and displays robust anti-tumor efficacy in vivo. Compared with that of AJ, the inclusion compound improved the therapeutic window and reduced toxicity by changing the injection time. Moreover, AJ-HP-β-CD showed a low dosage and long interval with less weight loss or other toxicities in vitro. These findings indicate the potential of AJ-HP-β-CD as a candidate anti-tumor agent for the treatment of kidney cancer.

## Figures and Tables

**Figure 1 molecules-25-05586-f001:**
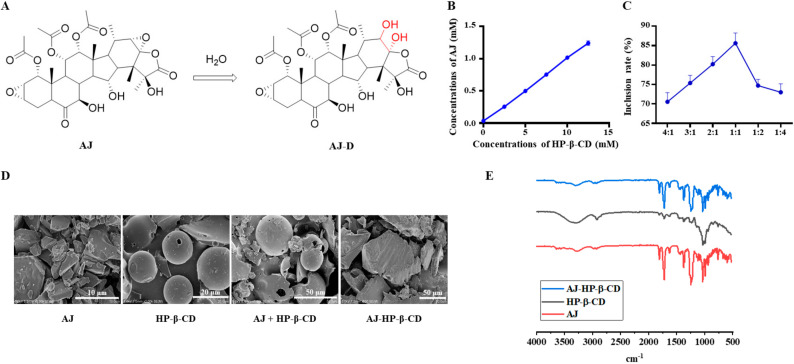
Preparation and characterization of taccalonolide AJ-hydroxypropyl-β-cyclodextrin (AJ-HP-β-CD). (**A**): The major hydrolysis compound of AJ. (**B**): The phase solubility diagram of AJ at 27 °C; (**C**): Molar ratio of AJ:HP-β-CD, Data are expressed as the mean ± standard deviation (SD). Statistical analysis was calculated using Student’s *t*-test (*n* = 3). (**D**): SEM images of AJ, HP-β-CD, physical mixture, and AJ-HP-β-CD. (**E**): FT-IR diagram of AJ (red), HP-β-CD (black), and AJ-HP-β-CD (blue).

**Figure 2 molecules-25-05586-f002:**
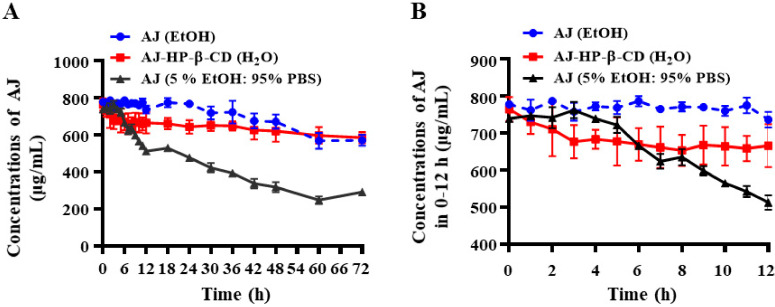
Stability of AJ and AJ-HP-β-CD. (**A**): The stability of AJ in EtOH (blue), AJ-HP-β-CD in water (red), and AJ in EtOH or 5% EtOH: 95% phosphate-buffered saline (PBS) (grey) at room temperature in comparison to free AJ by HPLC-evaporative light scattering detector (ELSD) (mean ± SD, *n*  =  3). (**B**): The stability of AJ in EtOH, AJ-HP-β-CD in water, and AJ in EtOH or 5% EtOH: 95% PBS during 0–12 h (*n* = 4). Statistical analysis was calculated using two-way ANOVA with Tukey’s post hoc test.

**Figure 3 molecules-25-05586-f003:**
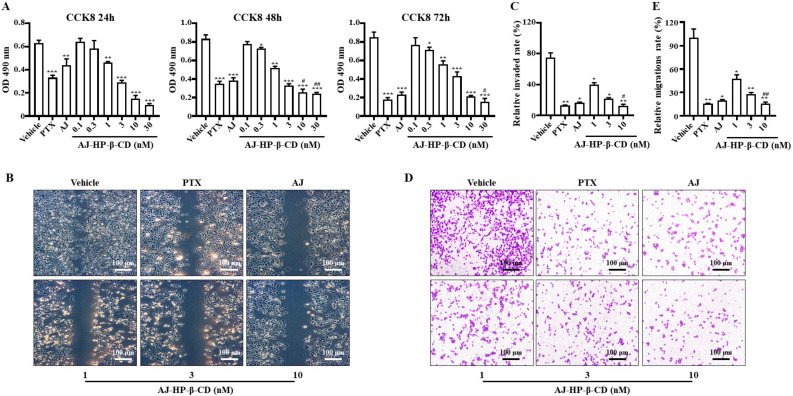
Effect of AJ-HP-β-CD on 786-O cell invasion and migration. (**A**): The cytotoxicity of AJ-HP-β-CD on 786-O cells with PTX (10 nM), AJ (10 nM), and AJ-HP-β-CD (0.3, 1, 3, 10, 30 nM) after 24, 48 and 72 h. (**B**), (**C**) Representative images scratch assay of 786-O cells with PTX (10 nM), AJ (10 nM), and AJ-HP-β-CD (1, 3, 10 nM) after 12 h. (**D**,**E**) Representative images of the transwell migration assay of 786-O cells with PTX (10 nM), AJ (10 nM), and AJ-HP-β-CD (1, 3, 10 nM) after 12 h. * *p* < 0.05, ** *p* < 0.01, *** *p* < 0.001 compared with vehicle (*n* = 3), ^#^
*p* < 0.05, ^##^
*p* < 0.01 compared to AJ, statistical analysis was calculated using Student’s *t*-test (*n* = 3).

**Figure 4 molecules-25-05586-f004:**
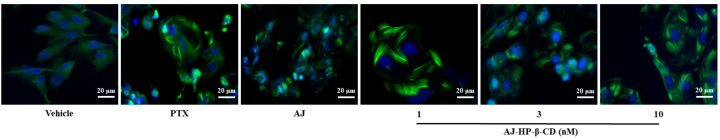
Effect of AJ-HP-β-CD on microtubules in 786-O cells. Images were overlaid electronically after cells were examined by fluorescence microscopy, and representative pictures were taken after 24 h. PTX (10 nM), AJ (10 nM), and AJ-HP-β-CD (1, 3, 10 nM).

**Figure 5 molecules-25-05586-f005:**
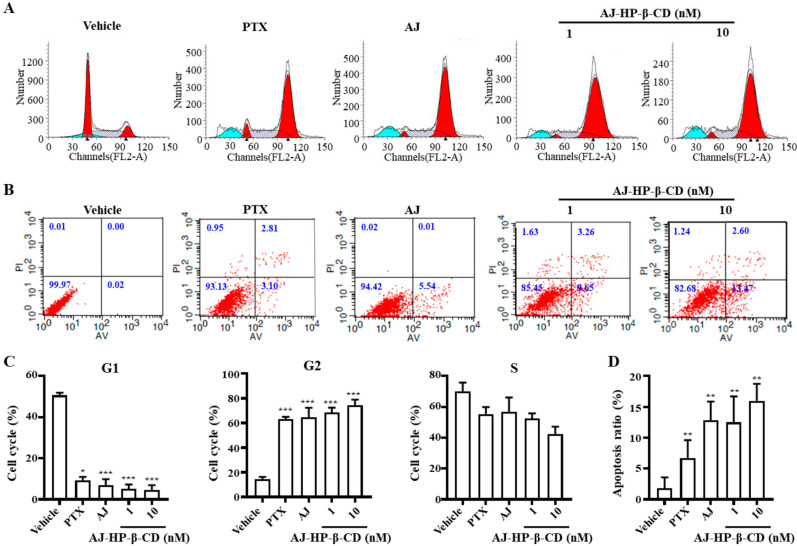
Effect of AJ-HP-β-CD on the cell cycle and apoptosis in 786-O cells. (**A**) 786-O cells were treated with PTX (10 nM), AJ (10 nM), and AJ-HP-β-CD (1, 3, 10 nM) for 12 h, and a flow cytometry analysis of the cell cycle assay was performed. (**B**) 786-O cells were treated with PTX (10 nM), AJ (10 nM), and AJ-HP-β-CD (1, 3, 10 nM) for 24 h. A flow cytometry analysis of apoptosis was performed in each group of cells. (**C**) The percentage of cells in G1, G2, S phases are shown in the histogram. (**D**) The percentage of apoptotic cells are quantitatively shown in the histogram, * *p* < 0.05, ** *p* < 0.01, *** *p* < 0.001 compared with vehicle (*n* = 3), statistical analysis was calculated using student’s *t*-test (*n* = 3).

**Figure 6 molecules-25-05586-f006:**
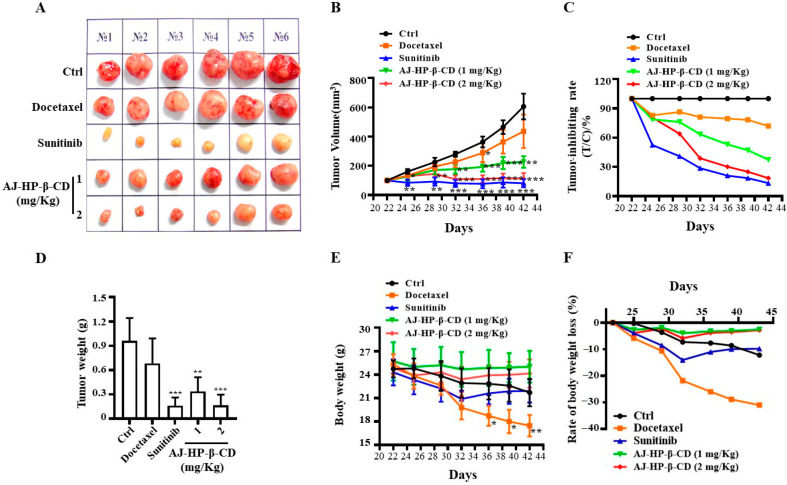
Effect of AJ-HP-β-CD on 786-O tumor growth in nude mice. Tumor diameter (**A**) was measured with a Vernier caliper, and the relative tumor volume (**B**) and tumor inhibition rate (**C**) were calculated. Tumor weight (**D**) and body weight (**E**) were measured, and the rate of body weight loss (**F**) was calculated. * *p* < 0.05, ** *p* < 0.01, and *** *p* < 0.005 compared with the control group, B and E statistical analysis was calculated using two-way ANOVA with Tukey’s post hoc test (*n* = 6), (**A**,**C**,**F**) (*n* = 6), (**D**) statistical analysis was calculated using student’s *t*-test (*n* = 6).

**Table 1 molecules-25-05586-t001:** Anti-proliferative profiles of TB, AJ, paclitaxel (PTX), and AJ-HP-β-CD in various human cancer cell lines.

Cell Line	Origin	IC_50_ (nM)			
		Taccalonolide B	Taccalonolide AJ	PTX	AJ-HP-β-CD
A2780	Epithelial ovarian cancer	>30,000	6.18 ± 0.446	1.87 ± 0.37	2.06 ± 0.13
786-O	Primary clear renal cell carcinoma	26,100.27	16.01 ± 1.54	4.85 ± 0.46	3.51 ± 0.79
ACHN	Metastatic papillary renal cell carcinoma	>30,000	35.96 ± 1.08	5.57 ± 0.28	9.82 ± 1.02
HepG2	Hepatocellular carcinoma	>30,000	9.49 ± 0.64	2.94 ± 0.25	1.90 ± 0.23
Hep-3B	Hepatocellular carcinoma	>30,000	8.36 ± 0.28	4.50 ± 0.57	1.90 ± 0.06
Huh-7	Hepatocellular carcinoma	>30,000	9.96 ± 0.21	5.18 ± 0.03	3.09 ± 0.20
HCT116	Human colorectal carcinoma	>30,000	14.62 ± 1.13	3.83 ± 0.35	4.24 ± 0.51
MCF-7	Human breast carcinoma	>30,000	23.87 ± 1.71	3.41 ± 0.69	16.54 ± 0.49

**Table 2 molecules-25-05586-t002:** Tumor volume of mice (mm^3^).

Group	Ctrl	Docetaxel, 10, QW	Sunitinib,	AJ-HP-β-CD	AJ-HP-β-CD
			80→40, QD	(1 mg/Kg, QW)	(2 mg/Kg, QW)
Day22	100.00 ± 0.00	100.00 ± 0.00	100.00 ± 0.00	100.00 ± 0.00	100.00 ± 0.00
Day25	160.76 ± 15.35	133.20 ± 13.34	84.30 ± 21.76 **	126.70 ± 29.78	129.69 ± 24.15
Day29	226.117 ± 26.82	195.30 ± 22.50	92.96 ± 27.57 **	171.30 ± 30.29 **	144.59 ± 20.81
Day32	279.26 ± 16.41	226.60 ± 36.36	80.60 ± 24.80 ***	177.90 ± 31.69 ***	108.59 ± 25.76 **
Day36	363.38 ± 34.04	288.90 ± 64.19 *	76.40 ± 30.65 ***	192.31 ± 32.03 ***	109.99 ± 25.49 ***
Day39	464.99 ± 42.54	364.20 ± 80.19	86.19 ± 35.07 ***	218.52 ± 41.42 ***	116.39 ± 30.53 ***
Day42	606.04 ± 80.43	435.60 ± 114.31	80.18 ± 26.90 ***	226.30 ± 39.39 ***	112.39 ± 38.61 ***

* *p* < 0.05, ** *p* < 0.01, and *** *p* < 0.005 compared with the control group, statistical analysis was calculated using two-way ANOVA with Tukey’s post hoc test (*n* = 6)

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
