# Peer review of "Development of Taccalonolide AJ-Hydroxypropyl-β-Cyclodextrin Inclusion Complexes for Treatment of Clear Cell Renal-Cell Carcinoma"

_molecules, 2020, doi:10.3390/molecules25235586_

Round 1

Reviewer 1 Report

All my previous comments are properly fixed. I recommend this revised version for publication. However, a minor correction needed.

Minor correction:

Line 54, 4.2 Nm should be changed to 4.2 nM

Author Response

Response 1: Thank for your kind reminder, and correct the “4.2 Nm” into “4.2 nM” in line 55.

Reviewer 2 Report

The manuscript “Development of Taccalonolide AJ-hydroxypropyl-B-cyclodextrin inclusion complexes for treatment of clear cell renal-cell carcinoma” by Han et al describe the formulation and biological activities of a cyclodextrin-based taccalonolide AJ.

The main goal of this manuscript is to identify a taccalonolide AJ formulation with an improved antitumor efficacy. Unfortunately, after all of the characterization of the cyclodextrin-AJ complex in vitro, it appears that it was administered via intratumoral injections in order to observe antitumor efficacy. This has been demonstrated for unconjugated taccalonolide AJ so it is unclear whether there is any advantage of this compound.

Furthermore, the drugs are administered by different routes for the toxicity (IV), PK (gavage for AJ and IV for the cyclodextrin conjugate), tissue distribution (gavage for AJ and IV for the cyclodextrin conjugate), and antitumor studies (intratumoral injection). This makes the data comparing these parameters to one another and particularly making comparisons between AJ and the cyclodextrin formulation that are administered by different routes impossible to interpret.

Claims are also made that the cyclodextrin formulation improves water solubility and chemical instability, but these comparisons are made with AJ in 5% tween-80, which curious as tween-80 is not used in the formulation of this compound previously. It is unclear why tween was added to this evaluation. It is suggested that a 5% EtOH, 95% PBS is an ‘unsuitable vehicle’, but this is a vehicle in which anticancer drugs are administered both in animal models and human patients. The fact that taccalonolide can be administered at its MTD in this aqueous solvent demonstrates that solubility is not a major concern for this compound (due to its potency so that not much drug is required to be in solution). There is also a statement made in the introduction that the poor water solubility of the compound is due to the C22,23 epoxide with no reference to this statement.

The major limitation of taccalonolide AJ for anticancer efficacy is a narrow therapeutic window that requires direct delivery to the tumor site, which has not been improved with the current formulation since the cyclodextrin formulation was also administered by intratumoral injection to observe antitumor efficacy. Therefore, it is unclear that this new formulation has any advantage as compared to the drug alone.

Other issues:
The duration of drug addition in the in vitro experiments in figures 3 – 5 should be included in the figure legend. There is a discrepancy in timing that is indicated between the text in section 2.6 that indicates that the flow cytometry experiments were performed 12 h after drug addition and in the methods section that indicates 24 h of drug addition.

It is unclear what the CCK-8 assay contributes in addition to the MTS assay. It is suggested that the CCK-8 assay is used as a measure of cytotoxicity but then the data are described as reductions in cell proliferation.

The description of the microtubule structures observed by immunofluorescence as rounded with multiple nuclei are inconsistent with the images shown in Figure 4 that demonstrate microtubule bundling in interphase cells.

The results of the flow-based apoptosis assay are poorly described and the axes are not well labeled to be able to interpret the data. How is the PI used differently to measure cell cycle in Figure 5A as compared to its incorporation as a measure of apoptosis in Figure 5B? Also, in figure 5D, the ‘rate of apoptosis’ is not being measured as this would require measurement over multiple time points. The y-axis of this graph should be relabeled.

The statistical test used for each analysis should be indicated in the figure legend. The methods section indicates that t-tests and ANOVAs are used, but it is not clear which analysis was done for which data.
The solubility data in Stable 1 is listed simply as a yes/no measure with no additional information provided, making this table uninterpretable

The source of taccalonolide AJ should be included in the methods section.

Author Response

Point 1: The main goal of this manuscript is to identify a taccalonolide AJ formulation with an improved antitumor efficacy. Unfortunately, after all of the characterization of the cyclodextrin-AJ complex in vitro, it appears that it was administered via intratumoral injections in order to observe antitumor efficacy. This has been demonstrated for unconjugated taccalonolide AJ so it is unclear whether there is any advantage of this compound. 

Response 1: Thank you for your valuable comments. Previous paper reported that taccalonolide AJ (AJ) has anti-tumor activity in vitro. Unfortunately, when AJ uses 5% EtOH: 95% PBS as solvent, it showed no therapeutic window for antitumor activity in vivo (reference 6). In our work, we use intravenous injection rather than intratumoral injection to observe antitumor efficacy. AJ-HP-β-CD formulation showed significant anti-tumor activity with less weight loss in vivo (Figure 6). We think this advanced unconjugated AJ to the follow-up research for possible drug development, which is ongoing now in our lab. The intratumoral injection is designed for establishing a tumor xenografts model, not for the administration of compound and its formulation. We clarify this in the section 4.11.

Point 2: Furthermore, the drugs are administered by different routes for the toxicity (IV), PK (gavage for AJ and IV for the cyclodextrin conjugate), tissue distribution (gavage for AJ and IV for the cyclodextrin conjugate), and antitumor studies (intratumoral injection). This makes the data comparing these parameters to one another and particularly making comparisons between AJ and the cyclodextrin formulation that are administered by different routes impossible to interpret.

Response 2: Thank you for your valuable comments. We used tail intravenous administration for toxicity and efficacy studies. And we used the comparison between oral and intravenous administration for PK and distribution studies. We think these administration methods are most common methods to learn the essential drugability of a lead compound or its formulation. Please also refer to our response one for the clarification of using intratumoral injection.

Point 3: Claims are also made that the cyclodextrin formulation improves water solubility and chemical instability, but these comparisons are made with AJ in 5% tween-80, which curious as tween-80 is not used in the formulation of this compound previously. It is unclear why tween was added to this evaluation. It is suggested that a 5% EtOH, 95% PBS is an ‘unsuitable vehicle’, but this is a vehicle in which anticancer drugs are administered both in animal models and human patients. The fact that taccalonolide can be administered at its MTD in this aqueous solvent demonstrates that solubility is not a major concern for this compound (due to its potency so that not much drug is required to be in solution). There is also a statement made in the introduction that the poor water solubility of the compound is due to the C22,23 epoxide with no reference to this statement.

Response 3: Thank you for pointing out this mistake. After reviewing the manuscript, we found herein the use of Tween-80 is a typo. We have corrected it to “5% EtOH and 95% PBS solution” in Figure 2.

Point 4: The major limitation of taccalonolide AJ for anticancer efficacy is a narrow therapeutic window that requires direct delivery to the tumor site, which has not been improved with the current formulation since the cyclodextrin formulation was also administered by intratumoral injection to observe antitumor efficacy. Therefore, it is unclear that this new formulation has any advantage as compared to the drug alone.

Response 4: Thank you for your valuable comments. Please also refer to our response 1 for the clarification of using intratumoral injection. Our work found that the cyclodextrin formulation showed significant antitumor efficacy by intravenous administration, which is the advantage compared to the drug alone.

Point 5: The duration of drug addition in the in vitro experiments in figures 3 – 5 should be included in the figure legend. There is a discrepancy in timing that is indicated between the text in section 2.6 that indicates that the flow cytometry experiments were performed 12 h after drug addition and in the methods section that indicates 24 h of drug addition.

Response 5: Revised as requested.

Point 6: It is unclear what the CCK-8 assay contributes in addition to the MTS assay. It is suggested that the CCK-8 assay is used as a measure of cytotoxicity but then the data are described as reductions in cell proliferation.

Response 6: The instructions manual suggests that the CCK-8 assay kit cannot only used for cytotoxicity, but also can measure cell proliferation. For example, these following literatures used CCK8 assay kit for this purpose (DOI: 10.1186/s12943-017-0685-9, 10.18632/aging.103391).

Point 7: The description of the microtubule structures observed by immunofluorescence as rounded with multiple nuclei are inconsistent with the images shown in Figure 4 that demonstrate microtubule bundling in interphase cells.

Response 7: Revised as requested. We have corrected the description of the microtubule structures to “The microtubule in AJ-HP-β-CD treated cells consistently appear to fill more of the cytoplasm and seem to be quite short in length.”

Point 8: The results of the flow-based apoptosis assay are poorly described and the axes are not well labeled to be able to interpret the data. How is the PI used differently to measure cell cycle in Figure 5A as compared to its incorporation as a measure of apoptosis in Figure 5B? Also, in figure 5D, the ‘rate of apoptosis’ is not being measured as this would require measurement over multiple time points. The y-axis of this graph should be relabeled.

Response 8: Thank you for your valuable comments. The results of flow-based apoptosis assay are further described in section 2.6. The axis of this graph is relabelled. Apoptosis was detected using an annexin V-FITC/PI apoptosis detection kit (BD Biosciences Pharmingen, San Diego, CA, USA), according to the manufacture’s protocol; the cell cycle analysis was performed with cell cycle detection kit (KenGen Biotech) according to the manufacture’s protocol. And the details of content were added in section 4.11. Admittedly studying the ‘rate of apoptosis’ over multiple time points is very important, in our study, we just measure the apoptosis rate of 786-O cells with drugs after 24h. And the y-axis of this graph was relabelled.

Point 9: The statistical test used for each analysis should be indicated in the figure legend. The methods section indicates that t-tests and ANOVAs are used, but it is not clear which analysis was done for which data.

Response 9: Revised as requested.

Point 10: The solubility data in Stable 1 is listed simply as a yes/no measure with no additional information provided, making this table uninterpretable.

Response 10: Revised as requested. We added the solubility information to Table S1.

Point 11: The source of taccalonolide AJ should be included in the methods section.

Response 11: Revised as requested. AJ was synthesized according to previous procedures, which was added to the section 4.2 (Page 9, Line 283).

Reviewer 3 Report

The manuscript “Development of cyclodextrin-based taccalonolide AJ complex as a novel tubulin stabilizer for clear cell renal-cell carcinoma” by Han et al, describes an interesting finding regarding the treatment of RCC; indeed the formulation of a cyclodextrin-based taccalonolide AJ, which maintains the anti-tumour effect following systemic injection, offers interesting perspective under the therapeutic point of view. However the manuscript presents some shortcomings in the presentation of data and in the clarity of methodologies. Further somewhere the author makes some conclusions that do not perfectly reflect the data presented. Therefore, I recommend that a major revision is necessary. My concerns are detailed below.

Comment 1: In row 193 you wrote “The fact that measurable anti-tumour activities were observed at the LD50 and 
MTD demonstrates that AJ-HP-β-CD does have a therapeutic window for anti-tumour activity (STable S2)”, how do you arrive at this conclusion? I believe to understand the point but the tableS2 shows only the mortality rate of the animals but not the anti-tumour activities.

Comment 2: In row 207 you wrote that docetaxel inhibited significantly the tumour growth, but it is true only at 36 days and the overall reduction of tumour volume is not comparable to the other drugs. Further the statistic is not proper visible in the graph. I suggest to add a summary table.

Comment 3: Figure 6C, specify how do calculate tumour-inhibiting rate. It’s not clear.

Comment 4: Figure 3B-C, how do you calculate the relative invaded rate? You reported for reference the number 25, but in this study I didn’t find an analogues experiment. In martial and method you wrote that you counted the number of cells, but I don’t understand that point, usually in this kind of experiment the width of the wound is mesured.

Comment 5: The title of paragraph 2.5 is “AJ-HP-β-CD enhances microtubule polymerization in 786-O cells “, how do you conclude it? You showed a figure in which it is visible an accumulation of tubulin following treatment, but to say that drugs have an effect on MT polymerization or length you have to perform aa polymerization assay (with purified tubulin for example), otherwise to have an idea of the consequences of treatments on MT growth/length you have to perform, for exemple, MT growth assay. Considering that the mechanism of action of MSAs, such as AJ, is to promote polymerization of tubulin and stabilize the polymer, I think it is necessary to make this point clearer for better charactherize AJ-HP-β-CD functioning.

Comment 6: In the discussion you conclude that the toxicity of AJ-HP-β-CD is 20 times reduced by adjusting the injections from once every three days to once per week. I can’t see in your experiments a comparison between these two different time points in terms of body weight.

Suggestions/corrections:

  1. Row 192: specify in the text that the dose tolerance test was made in acute.
  2. In supplementary materials in acute toxicity study correct “two doses” with “three doses”
  3. Figure 1B: specify X asses
  4. Row 133: TB, write in full the abbreviation
  5. Rows 146-147: “AJ-HP-β-CD in the 10 and 30 nM inhibitors 147 significantly compared with the AJ groups at 24, 48, and 72 h (both p < 0.01)”; could you write better this point? I suppose that there are some missing words in the statement.
  6. In rows 217-218 you wrote “The sunitinib and docetaxel groups exhibited weight loss, weakness, and inactivity as well as the production of 
yellowish urine (Figure 6E-F)”, but in figure 6E-F are present only graphs referring to body weight and not the other parameters. You should specify that the data is not shown and not that are indicating in the figure.
  7. I suppose that the schedule of treatment that you used for docetaxel and sunitinib has been descried according to studies present in literature? Could you specify the references?
  8. At what age do you perform the described in vivo experiments?
  9. Specify in material and method how do you perform tissue distribution assay. In rows 199-201 you wrote that ”the tissue distribution data showed that AJ-HP-β-CD (2 mg/kg) administered by gavage led to higher kidney deposition (2016.67 ± 156.15 ng/g and 116.67 ± 10.79 ng/g, p < 0.05) compared with AJ (2 mg/kg, 1283.33 ± 98.91 ng/g and 19.33 ± 3.51 ng/g) after 5 min and 30 min (Figure S5)”; the higher kidney deposition is also visible 10 e 60min. I suggest describing better this result.

Author Response

The manuscript “Development of cyclodextrin-based taccalonolide AJ complex as a novel tubulin stabilizer for clear cell renal-cell carcinoma” by Han et al, describes an interesting finding regarding the treatment of RCC; indeed the formulation of a cyclodextrin-based taccalonolide AJ, which maintains the anti-tumour effect following systemic injection, offers interesting perspective under the therapeutic point of view. However the manuscript presents some shortcomings in the presentation of data and in the clarity of methodologies. Further somewhere the author makes some conclusions that do not perfectly reflect the data presented. Therefore, I recommend that a major revision is necessary. My concerns are detailed below.

We thank the reviewer and appreciate the comments.

Comment 1: In row 193 you wrote “The fact that measurable anti-tumour activities were observed at the LD50 and MTD demonstrates that AJ-HP-β-CD does have a therapeutic window for anti-tumour activity (STable S2)”, how do you arrive at this conclusion? I believe to understand the point but the tableS2 shows only the mortality rate of the animals but not the anti-tumour activities.

Response 1: Thank you for pointing out this mistake. The LD50 and MTD data in table S2 only showed the mortality rate of mice. We corrected this conclusion in paragraph 2.7.

Comment 2: In row 207 you wrote that docetaxel inhibited significantly the tumour growth, but it is true only at 36 days and the overall reduction of tumour volume is not comparable to the other drugs. Further the statistic is not proper visible in the graph. I suggest to add a summary table.

Response 2: Thank you for your valuable comments. We add the tumour growth and tumour volume tables (Table 2) in Page 7 line 248 as you suggested.

Comment 3: Figure 6C, specify how do calculate tumour-inhibiting rate. It’s not clear.

Response 3: Thank you for your valuable comments. We add the calculation formula in Page12 Line 415.

Comment 4: Figure 3B-C, how do you calculate the relative invaded rate? You reported for reference the number 25, but in this study, I didn’t find an analogues experiment. In martial and method you wrote that you counted the number of cells, but I don’t understand that point, usually in this kind of experiment the width of the wound is mesured.

Response 4: Thank you for your valuable comments. The wound area was photographed immediately at 0 (W0) and 12 h (W12) and relative invaded rate in each group was calculated as previous reported (doi: org/10.2147/IJN.S269630; org/10.1007/s00109-020-01995-8). We added the method and calculation formula in Page 11 Line 364.

Comment 5: The title of paragraph 2.5 is “AJ-HP-β-CD enhances microtubule polymerization in 786-O cells “, how do you conclude it? You showed a figure in which it is visible an accumulation of tubulin following treatment, but to say that drugs have an effect on MT polymerization or length you have to perform aa polymerization assay (with purified tubulin for example), otherwise to have an idea of the consequences of treatments on MT growth/length you have to perform, for exemple, MT growth assay. Considering that the mechanism of action of MSAs, such as AJ, is to promote polymerization of tubulin and stabilize the polymer, I think it is necessary to make this point clearer for better charactherize AJ-HP-β-CD functioning.

Response 5: Thank you very much for pointing out this mistake. To avoid unnecessary misunderstanding, we corrected the title of paragraph 2.5 as “AJ-HP-β-CD promote microtubule accumulation in 786-O cells”.

Comment 6: In the discussion you conclude that the toxicity of AJ-HP-β-CD is 20 times reduced by adjusting the injections from once every three days to once per week. I can’t see in your experiments a comparison between these two different time points in terms of body weight.

Response 6: Thank you for your valuable comments. Previous paper reported that AJ at the concentration of 0.5 mg/kg showed an average weight loss of greater than 10%, and 2 mice succumbed to toxicity on days 11 and 12 (reference 6), which indicated that once every three days injections of AJ showed unacceptable toxicity. In our work rats treated with AJ-HP-β-CD (5 mg/kg) showed a weight loss within 1–6 days and then increased after 6–8 days (Figure S8), suggested the administration cycle of AJ-HP-β-CD should be adjusted at least once a week. To avoid unacceptable toxicity, we did not do comparison experiment. Indeed, to accurate clarified the toxicity of drug, comparison between these two different time points in terms of body weight should be explored in the future experiment.

Suggestions/corrections:

1. Row 192: specify in the text that the dose tolerance test was made in acute.

Response: Thank you for pointing out this mistake. We revised in Page 6 Line 206.

2. In supplementary materials in acute toxicity study correct “two doses” with “three doses”

Response: Thank you for pointing out this mistake. We revised in supplementary materials.

3. Figure 1B: specify X asses

Response: Thank you for pointing out this mistake. We revised in figure 1B, Page 2 Line 106.

4. Row 133: TB, write in full the abbreviation

Response: Thank you for pointing out this mistake. We revised in Page 3 Line 141.

5. Rows 146-147: “AJ-HP-β-CD in the 10 and 30 nM inhibitors significantly compared with the AJ groups at 24, 48, and 72 h (both p < 0.01)”; couldl you write better this point? I suppose that there are some missing words in the statement.

Response: Thank you for pointing out this mistake. We revised in Page 4 Line 157.

6. In rows 217-218 you wrote “The sunitinib and docetaxel groups exhibited weight loss, weakness, and inactivity as well as the production of 
yellowish urine (Figure 6E-F)”, but in figure 6E-F are present only graphs referring to body weight and not the other parameters. You should specify that the data is not shown and not that are indicating in the figure.

Response: Thank you for pointing out this mistake. We revised as requested in Page 7 Line 235.

7. I suppose that the schedule of treatment that you used for docetaxel and sunitinib has been descried according to studies present in literature? Could you specify the references?

Response: Thank you for pointing out this mistake. We added the reference 6 and 22.

8. At what age do you perform the described in vivo experiments?

Response: Thank you for pointing out this mistake. We revised as requested in Page 11 Line 397.

9. Specify in material and method how do you perform tissue distribution assay. In rows 199-201 you wrote that ”the tissue distribution data showed that AJ-HP-β-CD (2 mg/kg) administered by gavage led to higher kidney deposition (2016.67 ± 156.15 ng/g and 116.67 ± 10.79 ng/g, p < 0.05) compared with AJ (2 mg/kg, 1283.33 ± 98.91 ng/g and 19.33 ± 3.51 ng/g) after 5 min and 30 min (Figure S5)”; the higher kidney deposition is also visible 10 e 60min. I suggest describing better this result.

Response: Thank you for pointing out this mistake. We revised in Page 6 Line 214.

Round 2

Reviewer 2 Report

While I would be very enthusiastic about the finding that a cyclodextrin complex of taccalonolide AJ has an improved therapeutic window for systemic antitumor efficacy, I have significant concerns with regard to the rigor of this manuscript, which I have now reviewed in at least three iterations.

While the reviewer responses suggest that each of the concerns has been addressed, many of the concerns have not actually been resolved or have been resolved very superficially. Several of these issues are indicated below, but this is not an exhaustive list.

  • The author’s state that no natural taccalonolide directly interacts with tubulin, however, taccalonolide AF was originally isolated from a natural source and was the first taccalonolide to demonstrate a direct ability to polymerize tubulin in biochemical preparations
  • The authors state that a 5% EtOH/95% PBS vehicle is ‘unsuitable’, but this vehicle is used for the clinical administration of other clinically used MTAs, such as eribulin
  • There is still no evidence clearly provided to the author’s statement of the main hydrolysis product proposed in Figure 1A. The only spectroscopic data in the supplemental information on AJ appears to be in MeOD (which is not defined)
  • The authors suggest that they have now included the source of the taccalonolide material used in this study by including a statement that the material was prepared following our previous procedures. However, the reference cited here does not contain any of the authors from this current manuscript.

However, my biggest concerns stem from the lack of rigor in the preparation of this manuscript, even upon multiple rejections and resubmissions. When issues have been brought up such as inconsistencies in the vehicles used or the routes of administration for in vivo studies, the authors indicate that these were ‘mistakes’ and simply corrected the language. While mistakes can indeed be made, the number of these is very worrisome as they suggest a significant lack of attention to detail in the preparation of this work that precludes the ability to review the actual science. The accidental inclusion of tween in the stability studies and saying that antitumor studies were done with intratumoral injection and then corrected to IV injections are major ‘typos’ that suggest a complete lack of attention to detail and bring into question every other experiment described in this manuscript.

Finally, it is indicated that much of the in vivo data described in the supplemental methods was performed in rats. However, the antitumor studies in the main body of the manuscript are described to be done in mice. This is concerning either from a scientific standpoint (there is a lack of consistency between in vivo experiments done in different species with no attempt to discuss the relationship of the mouse and rat studies to one another) or from an organizational standpoint (if this is yet another ‘typo’ I have little confidence in anything written in this manuscript).

The lack of rigor in the preparation of this manuscript, particularly after multiple resubmissions, leads to a complete lack of confidence in this study on my part and I am too skeptical of this work to recommend publication now or even after major revisions.

Author Response

While I would be very enthusiastic about the finding that a cyclodextrin complex of taccalonolide AJ has an improved therapeutic window for systemic antitumor efficacy, I have significant concerns with regard to the rigor of this manuscript, which I have now reviewed in at least three iterations.

While the reviewer responses suggest that each of the concerns has been addressed, many of the concerns have not actually been resolved or have been resolved very superficially. Several of these issues are indicated below, but this is not an exhaustive list.

  • The author’s state that no natural taccalonolide directly interacts with tubulin, however, taccalonolide AF was originally isolated from a natural source and was the first taccalonolide to demonstrate a direct ability to polymerize tubulin in biochemical preparations
  • The authors state that a 5% EtOH/95% PBS vehicle is ‘unsuitable’, but this vehicle is used for the clinical administration of other clinically used MTAs, such as eribulin
  • There is still no evidence clearly provided to the author’s statement of the main hydrolysis product proposed in Figure 1A. The only spectroscopic data in the supplemental information on AJ appears to be in MeOD (which is not defined)
  • The authors suggest that they have now included the source of the taccalonolide material used in this study by including a statement that the material was prepared following our previous procedures. However, the reference cited here does not contain any of the authors from this current manuscript.

However, my biggest concerns stem from the lack of rigor in the preparation of this manuscript, even upon multiple rejections and resubmissions. When issues have been brought up such as inconsistencies in the vehicles used or the routes of administration for in vivo studies, the authors indicate that these were ‘mistakes’ and simply corrected the language. While mistakes can indeed be made, the number of these is very worrisome as they suggest a significant lack of attention to detail in the preparation of this work that precludes the ability to review the actual science. The accidental inclusion of tween in the stability studies and saying that antitumor studies were done with intratumoral injection and then corrected to IV injections are major ‘typos’ that suggest a complete lack of attention to detail and bring into question every other experiment described in this manuscript.

Finally, it is indicated that much of the in vivo data described in the supplemental methods was performed in rats. However, the antitumor studies in the main body of the manuscript are described to be done in mice. This is concerning either from a scientific standpoint (there is a lack of consistency between in vivo experiments done in different species with no attempt to discuss the relationship of the mouse and rat studies to one another) or from an organizational standpoint (if this is yet another ‘typo’ I have little confidence in anything written in this manuscript).

The lack of rigor in the preparation of this manuscript, particularly after multiple resubmissions, leads to a complete lack of confidence in this study on my part and I am too skeptical of this work to recommend publication now or even after major revisions.

Thank you very much for your comments!

Reviewer 3 Report

The authors provided most of the corrections/suggestions of the first report; however there are some points that have still to be clarify:

Minor:

  • 229-231 “The results showed that the tumour volume was reduced and 
tumour growth was inhibited significantly in the docetaxel, sunitinib, and AJ-HP-β-CD groups 
compared with vehicle control animals (Figure 6A-B)”. 
Is not what the graph shows, indeed docetaxel shows only a tendency in tumour growth reduction. I appreciated the table 2 but you have to specify in the text the inclusion of table 2 otherwise the table is not contextualised.
  • 250-251 The sunitinib and docetaxel groups exhibited weight loss (Figure 6E-F), weakness, and inactivity as well as the production of yellowish urine (Figure 6E-F). The second brackets have to be removed. Do you have quantifiable data on inactivity and yellowish urine? If they not exist and this is only a qualitative analysis you have to write that the animal “appear” etc.
  • Figure 6C. I appreciated the formula to calculate tumour-inhibiting rate, but the control at this point is not 0 but 100%, the graph is not correct.
  • The reference number 26 is not correct for the scratch test.
  • 183 2.5. AJ-HP-β-CD promoteS microtubule accumulation in 786-O cells

Major: for the paragraph 2.5, you have changed the title of the paragraph making the description of the result “milder” and so acceptable but I really would have appreciated to see an experiment addressed to MT polymerisation upon drug treatment. I leave to the edhitor this consideration.

Author Response

The authors provided most of the corrections/suggestions of the first report; however, there are some points that have still to be clarify:

Minor:

Comment 1: 229-231 “The results showed that the tumour volume was reduced and tumour growth was inhibited significantly in the docetaxel, sunitinib, and AJ-HP-β-CD groups compared with vehicle control animals (Figure 6A-B)”. Is not what the graph shows, indeed docetaxel shows only a tendency in tumour growth reduction. I appreciated the table 2 but you have to specify in the text the inclusion of table 2 otherwise the table is not contextualized.

Response 1: Thank you for your valuable comments. We added the table 2 in 225-231.

Comment 2: 250-251 The sunitinib and docetaxel groups exhibited weight loss (Figure 6E-F), weakness, and inactivity as well as the production of yellowish urine (Figure 6E-F). The second brackets have to be removed. Do you have quantifiable data on inactivity and yellowish urine? If they not exist and this is only a qualitative analysis you have to write that the animal “appear” etc.

Response 2: Thank you for your valuable comments. We removed the “(Figure 6E-F)” in Line 236, and add the description “and the animals appeared…” in line 235 as you suggested.

Comment 3: Figure 6C. I appreciated the formula to calculate tumour-inhibiting rate, but the control at this point is not 0 but 100%, the graph is not correct.

Response 3: Thank you very much for pointing out this mistake. We corrected the graph in Figure 6C.

Comment 4: The reference number 26 is not correct for the scratch test.

Response 4: Thank you very much for pointing out this mistake. We corrected the reference 26 for the scratch test in paragraph 4.9.

Comment 5: 183 2.5. AJ-HP-β-CD promote S microtubule accumulation in 786-O cells

Response 5: Thank you for your valuable comments.

Major: for the paragraph 2.5, you have changed the title of the paragraph making the description of the result “milder” and so acceptable but I really would have appreciated to see an experiment addressed to MT polymerization upon drug treatment. I leave to the editor this consideration.

Response 6: We thank the reviewer and appreciate the comments. In our study we prepared a cyclodextrin-based carrier of AJ which mainly focus on the treatment effects of drugs; the underline mechanism was just examined by immunofluorescent staining of tubulin. Admittedly, the accurate mechanisms such as microtubule polymerization need to be explored. Unfortunately, after consulted several reagent vendors and found that the CytoDYNAMIX Screen kit (BK006P or BK011P, Cytoskeleton Inc., Denver, CO, USA) requires a relatively long delivery period (about a month), and this experiment cannot be completed within the specified time. Actually, we have conceived a series experiments to explore the possible mechanism of AJ-HP-β-CD in renal-cell carcinoma in the follow-up study which include the MT polymerization.

This manuscript is a resubmission of an earlier submission. The following is a list of the peer review reports and author responses from that submission.

Round 1

Reviewer 1 Report

The MS discusses about the development of cyclodextrin based Taccalonide AJ formulation for the treatment of ccRCC.  The strength of the MS is the data and the design of the experiments.  However, the MS is poorly written and has a number of fallacies.  As such, this version of the MS is not suitable for publication in any journal.

The figure legends are not a proper representation of the attached figures in Fig. 3, 4, 5, 7 and 8. The MS does not describe the figures 3B, 3D, 3E, 5E and 5F. Data in line 100, Page 3/19 describes Figure 3C rather than 3B. Line 111, Page 3/19. Authors mentioned a relatively narrow size distribution. However, they have no data to support this claim. The IR figure (4C) for AJ and HPbCD+AJ looks the same. Is it not expected that the complex has a different IR spectrum than the parental compound? The authors could have discussed more about the similarities and differences between the NMR spectra of AJ, HPbCD and complex rather than just mentioning the resonance peaks. Additionally, the NMR spectrum of HPbCD could have also been arranged in Fig. 4D to have a comparison. The authors have not specified if the IC50s mentioned in the Table 1 correspond to AJ or the complex. No discussion on the results of the scratch assay (5E-F) were mentioned in the text. The arrangement of the Fig. 5 is not in alphabetical order, leading to more confusion for the readers. No statistics were performed for the graphs in 7A and 7B. Additionally, wherever stats were performed and significant differences were mentioned, there were no explanations on what were the groups that were compared to. In line 218, page 10/19; Fig. 8C does not describe about body weights of the animals. Similarly, line 219, Fig. 8D does not represent the mean tumor weights. According to Figure 8A, there seems to be a significant error in the tumor volumes, especially in the control animals. However, the error bars in Fig. 8B and 8C for control animals remains very low.  Are these graphs a proper representation of the results? Additionally, with the very low error bars in Fig. 8B and 8C, one could argue that the green group (AJ 1 mg/kg) seems to be significantly different from that of controls. Is this assessment correct? Also, additional groups to include in the experiment are HPbCD alone and AJ alone (not in complex). Finally, the procedure for the xenografts lack experimental details on the route of administration, dosage regimen, dosage interval and the number of cancer cells injected per animal. No NMR data available in supplementary figures (S1, S2).

Reviewer 2 Report

The manuscript “Development of cyclodextrin-based taccalonolide AJ complex as a novel tubulin stabilizer for clear cell renal-cell carcinoma” by Han et al describe the formulation and biological activities of a cyclodextrin-based taccalonolide AJ. The finding that this formulation retains in vivo antitumor efficacy is significant as the native compound has previously only shown antitumor activity when directly injected into the tumor site due to a narrow therapeutic window and short serum half-life of less than 10 min. Unfortunately, my enthusiasm for this work is tempered by lack of attention to detail in its presentation as described in more detail below. Overall, the important finding that the cyclodextrin formulated AJ retains activity in vivo with systemic injection is important. However, the value of this finding is lost due to a lack of clarity in the presentation of the data and the fact that it is not properly described in context with the literature. This paper would be significantly strengthened by comparison of this new AJ formulation to AJ alone (as opposed to only the current comparison to PTX). Additionally, interpretation of this data requires a comparison of the serum half-life of AJ/HPBCD in comparison to AJ alone, particularly since these compounds appear to have a similar MTD in vivo.

One of the major rationales for formulating the cyclodextrin-based AJ formulation is stated as AJ itself having “low aqueous solubility”. However, the referenced papers deliver AJ in vivo by systemic injection at the maximum tolerated dose in 5-10% EtOH vehicle, which is possible because it is so potent (1 – 2 mg/kg as MTD). The cyclodextrin AJ formulation is administered in 5% EtOH: 5% Tween-80 so there is no functional improvement in aqueous solubility based on the solvent used for dosing and frankly no need to improve the solubility due to the low dosing required. The authors should instead focus on the low therapeutic window and short half-life of AJ as a rationale for formulation, not aqueous solubility. The manuscript would be significantly strengthened by measurements of serum half-life of the cyclodextrin formulation in comparison to AJ alone, which would directly demonstrate this.

The introduction needs to be carefully revisited both for accuracy and for clarity. The statement that PTX and carboplatin are ‘now’ in Phase II clinical trials is based on a 2009 reference. Suggesting that this is a current trial is likely not accurate. It is stated that PTX ‘preferentially’ binds to the β-subunit of tubulin, this suggests that there are multiple binding sites where it could bind, is this the intent? Both the introduction and the discussion require careful editing to clarify the rationale for the current study in the introduction and put the current findings into context of the current literature in the discussion – currently both sections lack clear overall focus and direction. The discussion should incorporate the chemical stability studies with the in vivo efficacy (and also PK) to more fully understand the role of the cyclodextrin formulation in increasing the therapeutic window.

There are several references that look to be wrong in the introduction. The ability of the taccalonolides to circumvent drug resistance mechanisms should reference the 2008 Cancer Research paper by Risinger et al. and the 2018 JNP paper by Ola et al that specifically address the efficacy of the taccalonolides in drug resistant in vivo models. The paper that is referenced here (#8, Risinger et al JNP 2017) does not describe drug resistance but does describe the pharmacokinetics of the taccalonolides. The reference #9 for AJ semisynthesis is not correct. The statement that the semisynthetic AJ can circumvent drug resistance by binding covalently to D226 should reference the Li et al JACS paper first describing AJ semisynthesis and the Risinger et al 2013 Cancer Research paper that first describes the covalent binding in addition to the crystal structure paper.  A large number of the references I checked were not correctly referenced and many were totally unrelated. This is a major issue in the discussion as well.

The source of plant material should be included in the methods section.

Figure 3 – the panel labels in the figure do not match up with the figure legend

Table 1 is titled as the anti-proliferative profile of the compounds, but in the text these results are referred to as cytotoxicity studies. Is the IC50 calculated as the concentration that inhibits 50% proliferation, causes 50% cytotoxicity, or gives a 50% maximal effect?  Errors for multiple IC50 values for each cell line/compound combination should be included in the table. Also, full curves of the growth inhibitory/cytotoxic activity would be optimal to include in order to evaluate both potency and efficacy. It is curious that the AJ/HPβCD formulation, which is the focus on this manuscript, is not included in this analysis, a direct comparison of this current formulation to AJ alone should be the focus on this work.

In figures 5 - 7, why is AJ/HPβCD compared to PTX but not to AJ alone? It is unclear what the interpretation of these experiments are other than AJ/HPβCD still retains AJ-like activity, but then is not compared directly to AJ alone.

The figure 7 legend does not include the correct concentrations of AJ/HPβCD tested in the figure.

In figure 7B, the ‘rate of apoptosis’ is not being measured as this would require measurement over multiple time points. The y-axis of this graph should be relabeled.

The statement that “AJ has distinct pharmacokinetic properties” on page 10 needs to be clarified, what specifically does this mean and how do the properties described in reference 8 lead to the current evaluations in vivo for AJ/HPβCD? The short serum half-life? Evaluating the in vivo activity and PK of AJ/HPβCD directly compared to AJ would significantly increase the impact of this study and greatly aid in the interpretation of the impact of this formulation.

On page 10, line 198 it suggests that mice are treated with AJ, is this supposed to read AJ/HPβCD? There seem to be multiple places where these two terms are not used correctly to describe the compound that was actually used.

For figure 8, what do the points and error bars represent? Mean and SEM?? This figure legend needs parts A, B, C, and D delineated. What is the difference between tumor volume and relative tumor volume in parts B and C? What day were the tumors harvested and shown in part A? Presumably this was on the last day on the graphs (43)? The range of sizes of the AJ/HPβCD tumors in A seem inconsistent with the small error bars shown in the graph. Bar graphs with individual tumor volumes and animal weights at least at the time of trial initiation and end of trial would be optimal to fully appreciate the biological response and variability.

The statement that there is no evidence to support the in vivo antitumor efficacy of AJ on pg 12, line 233 is not consistent with reference #8, where intratumoral injections caused robust and persistent antitumor efficacy. It is true that AJ does not have a therapeutic window for antitumor efficacy by systemic injection, which is proposed to be the advantage of the current formulation. However, the methods and figure legend lack any detail with regard to the route of administration of these compounds, which is critical to make this statement. I did find the route of administration in supplemental table 1 (under the incorrect heading ‘medication’) but this needs to be stated in the methods section.

It is unclear why there is a random paragraph on autophagy in the discussion. Autophagy is not evaluated in the manuscript at all. Then there is a statement that the study showed that the taccalonolides could inhibit VEGF and mTOR expression, but this data is not shown. Instead of discussing these areas of biology that have not been addressed at all in the manuscript, the authors should focus on the advantage of AJ/HPβCD for eliciting antitumor efficacy by systemic injection as compared to AJ alone and the rationale for why this is the case – drug stability/pharmacokinetics.

The equation for tumor volume calculation in the methods is incomplete

When more than two discrete values are being compared, as in 5D, H, F, an ANOVA should be used as opposed to a t-test.

More information on the statistical tests evaluated in figure 8 need to be described. If values are being compared between treatment groups and over time then a 2-way ANOVA should be used for statistical analysis.

Figures S1 and S2 in the supplemental data are missing.

Supplemental table 1 states that docetaxel was used in the antitumor trial, but paclitaxel is indicated in Figure 8.